# Hybrid Encrypted Watermarking Algorithm for Medical Images Based on DCT and Improved DarkNet53

**Dekai Li [1], Jingbing Li [1,*], Uzair Aslam Bhatti [1], Saqib Ali Nawaz [1], Jing Liu [2], Yen-Wei Chen [3] and Lei Cao [1]**

[1]  School of Information and Communication Engineering, Hainan University, Haikou 570228, China
[2]  Research Center for Healthcare Data Science, Zhejiang Laboratory, Hangzhou 311121, China
[3]  Graduate School of Information Science and Engineering, Ritsumeikan University,
    Kusatsu 525-8577, Shiga, Japan
*  Correspondence: jingbingli2008@hotmail.com; Tel.: +86-136-3765-8206

**Abstract:** To solve the problem of robustness of encrypted medical image watermarking algorithms, a zero watermarking algorithm based on the discrete cosine transform (DCT) and an improved DarkNet53 convolutional neural network is proposed. The algorithm targets medical images in the encrypted domain. In this algorithm, DCT is performed on the encrypted medical image to extract 32-bit features as feature 1. DarkNet53, a pre-trained network, was chosen for migration learning for the network model. The network uses a fully connected layer and a regression layer instead of the original Softmax layer and classification layer, changing the original classification network into a regression network with an output of 128. With these transformations, 128-bit features can be extracted from encrypted medical images by this network, and then DCT is performed to extract 32-bit features as feature 2. The fusion of features 1 and 2 can effectively improve the robustness of the algorithm. The experimental results show that the algorithm can accurately distinguish different encrypted medical images and can effectively restore the original information from the encrypted watermarked information under traditional and geometric attacks. Compared with other algorithms, the proposed method demonstrates better robustness and invisibility.

**Keywords:** cryptographic medical images; convolutional neural network; DarkNet53; migration learning; DCT

## 1. Introduction

With the gradual digitization of current medical technology, a large amount of medical information needs to be transmitted via the internet. Watermarking technology is effective for concealing patient information in images and facilitating transmission. However, this puts the patient's information at risk of leakage. The use of digital image watermarking technology in the medical industry is steadily becoming more widespread [1,2], thanks to the rapid advancements that have been made in artificial intelligence, computer vision, image processing, and other areas of study. At this time, the vast majority of patient information, including images of patients, must be transmitted online. Researchers must devise a method to safeguard the patients' information and prevent it from being stolen. The digital watermarking technique offers a potentially useful solution to the problem described above [3]. This is because the technology is undetectable, and it also has the capacity to continually upgrade and improve upon older encryption methods. Because of this capability, sensitive patient information can be concealed within medical images.

Since the unique qualities of medical images mean that the watermark will not interfere with the doctor's ability to diagnose the original image, zero watermark technology is an excellent solution to this issue [4]. Currently, most digital picture watermarking methods are designed for use in the plaintext domain, where they can be embedded and later extracted. However, the patient's private information could be stolen if the medical

image in the plaintext domain is intercepted in transit. Therefore, the original image cannot be guaranteed to be secure using the digital watermarking process in the plaintext domain, especially when transmitting medical images. It has been shown that the digital watermarking approach in the ciphertext domain is superior at resolving the aforementioned issue [5].

By embedding and extracting the watermark in the ciphertext domain, the information carried by the carrier image will be effectively hidden, significantly improving the carrier image's security. Moreover, homomorphic encryption can safely hand over the encrypted carrier image and watermark to a third party for processing, so there is no need to worry about security risks such as information theft and alteration [6]. The zero watermarking technique is implemented by taking advantage of the vital resources of third parties. The original image needs to be encrypted first for embedding and extracting watermarks in the ciphertext domain. Researchers have provided a large number of image encryption algorithms, such as Yang et al.'s proposed image encryption algorithm based on adaptive two-dimensional compression perception and a chaotic system, which improves the visual security of encrypted images and can effectively enhance the embedding rate [7]. Musanna et al. proposed an image encryption algorithm based on fractional chaos and cellular neural networks [8]. Zhong et al. proposed a multi-image encryption algorithm based on wavelet transform and 3D shuffle scrambling by performing a wavelet transform on each layer of the reconstructed image cube and then using a 3D shuffle algorithm and heteroskedastic operation to achieve encryption, achieving high operation speed and resistance to attack [9]. Kamil et al. [10] proposed block-wise reversible watermarking technique for security of images using dynamic reversible blocks. The work of Sahu et al. [11] shows the significance of tools used for the security of data from tempering and highlights the forensic techniques to further improve data security. While previous research has focused on securing 2D mesh fog data, the work of Raghunandan et al. [12] presents an innovative method of securing 3D point fog data. Initially, the sequence produced by the chaotic behavior is used to transform the coordinates of the fog data. The expanded scope of the suggested map is then represented via bifurcation analysis. Then, the Lyapunov exponent and the approximate entropy are used to evaluate the proposed chaotic system.

There are two basic categories for watermark embedding: spatial domain and transform domain [13]. Medical image watermarking strategies are typically studied in the transform domain because it is more challenging to demonstrate improved resilience for spatial domain embedding and extraction procedures. To implement transform domain-based image watermarking, several transform techniques are applied to the carrier image to extract the transform coefficients, which are then modified to embed the watermark. Conventional transform techniques include the discrete cosine transform (DCT), the discrete wavelet transform (DWT), the singular value decomposition (SVD), etc. [14–16]. A technique for zero watermarking images using DCT and DFT was proposed by S. Xing et al. [17]. To create the zero-watermark image, we first perform a discrete Fourier transformation (DFT) on the image to obtain the coefficient matrix, then a discrete cosine transform (DCT) to select the low-frequency coefficients as the feature image of the original image, and finally, an exclusive-or (XOR) operation on the encrypted watermark data. Using 2D discrete wavelet transforms, SVDs, and chaotic maps, Wang Kunshu et al. [18] proposed a safe method of watermarking two-color images. First, a color space transformation (RGB to NTSC) is performed on the original image and the encrypted watermark, and then a multi-level 2D discrete wavelet transform (DWT) is performed. Ultimately, the encrypted watermark is embedded by altering the single values of the original image's low-frequency sub-bands.

The expanding applications of deep learning have led to its increasing popularity among researchers as the go-to method for addressing complex problems [19]. Deep learning shows excellent performance even in computer vision tasks such as pedestrian recognition, image categorization, etc. [20,21]. As one of the most popular network models, convolutional neural networks leverage their formidable processing capability to reliably

extract deep information. Recently, Liu et al. [22] suggested an undetectable and robust watermarking approach employing convolutional neural networks to address the shortcomings of digital watermarking techniques in the face of geometric attacks. Zero-watermarking algorithms for medical images using the VGG-19 deep convolutional neural network were proposed by Han et al. [23]. The algorithm first extracts deep features from medical images using a pre-trained VGG19 network, then uses the Fourier transform on the extracted features, selects the transformed 64-bit low-frequency coefficients to construct the feature matrix, then uses the hash transform to generate a binary sequence, and finally uses the encrypted watermarked image and the binary sequence to perform calculations to achieve zero watermarking. When compared to conventional algorithms, this one performs better in geometric attacks.

In summary, although many watermarking algorithms have been studied, watermarking algorithms for encrypted medical images are still inadequate, and even fewer algorithms can achieve better robustness, especially against geometric attacks. Therefore, in this paper, we propose an encrypted medical image watermarking algorithm based on DCT and an improved DarkNet53 convolutional neural network, which has the primary purpose of authentication and privacy information protection [24]. Firstly, DarkNet53's pre-trained network needs to be improved and trained. The medical images are then encrypted, and the encrypted medical images are subjected to DCT to extract features 1 and 2. At the same time, the encrypted medical image is fed into the improved DarkNet53 convolutional neural network to extract feature 2. Secondly, the watermarked image is encrypted using chaos encryption. Finally, features 1 and 2 are simply fused, and an aliasing operation is performed on the encrypted watermark to achieve watermark embedding and extraction.

The main contributions of this study are as follows:

(1)  It is proposed that DCT and an improved DarkNet53 convolutional neural network can be used to make a robust zero-watermarking algorithm for cryptographic medical images;

(2)  Encrypting both the carrier image and the watermark information ensures that the carrier image information is safe and that the watermark information is safe and easy to see;

(3)  The network's structure is changed and trained with a certain set of data so that robust features can be extracted;

(4)  The algorithm has high robustness against both geometric and conventional attacks.

## 2. Basic Theory

### 2.1. DarkNet53 Convolutional Neural Network

DarkNet53 is the backbone feature extraction network used by the target detection network YOLOv3 for extracting features with 8, 16, and 32-fold downsampling, respectively [25]. The network structure of DarkNet53 is shown in Figure 1, and this network model combines the deep residual network with DarkNet19, the feature extraction network used by YOLOv2 [26]. This network partly makes extensive use of $1 \times 1$ convolution and $3 \times 3$ convolution, where $1 \times 1$ is mainly applied to the expansion and reduction of channels. The overall convolutional network uses a structural model of a convolutional layer + batch normalization (BN) layer + Leaky ReLU layer.

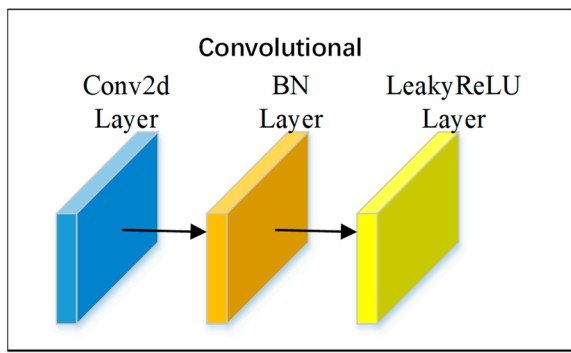

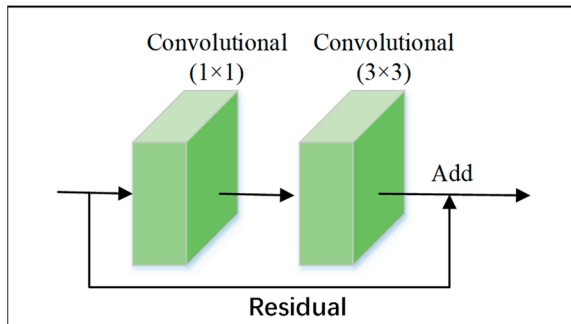

**Figure 1.** Structure of DarkNet53 network model.

## 2.2. Tent Map

The tent map is a segmented linear mapping in mathematics with a tent-like function image, as shown in Figure 2 [27]. It is also a two-dimensional chaotic mapping, which is widely used in chaotic cryptosystems (e.g., image encryption) and is often used in the generation of chaotic spreading codes, in the construction of chaotic cryptosystems, and in the implementation of chaotic preference algorithms. The chaotic sequences generated by its mapping have good statistical properties. The formula is as follows:

$$X_n = \begin{cases} \dfrac{X_n}{\alpha}, 0 \le X_n < \alpha \\ \dfrac{1 - X_n}{1 - \alpha}, \alpha \le X_n < 1 \end{cases} \quad (1)$$

The mapping is in a chaotic state when $\alpha \in (0, 1)$ and has a uniform distribution function on (0, 1).

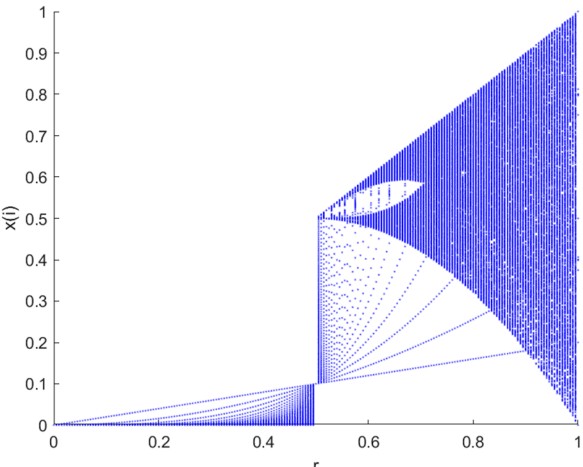

**Figure 2.** Tent map bifurcation diagram [28].

### 2.3. Discrete Cosine Transform (DCT)

A commonly used one-dimensional DCT transformation formula is as follows:

$$F(u) = c(u) \sum_{i=0}^{N-1} f(i) \cos\left[\frac{(i+0.5)\pi}{N} u\right] \tag{2}$$

Among them, $c(u) = \begin{cases} \sqrt{\dfrac{1}{N}}, u=0 \\ \sqrt{\dfrac{1}{N}}, u \neq 0 \end{cases}$ , $c(u)$ is a coefficient and N is the total number of $f(x)$.

The two-dimensional discrete cosine transform formula is as follows:

$$F(u,v) = C(u)C(v) \sum_{x=0}^{M-1} \sum_{y=0}^{N-1} f(x,y) \cos\left[\frac{(x+0.5)u\pi}{M}\right] \cos\left[\frac{(y+0.5)v\pi}{N}\right] \tag{3}$$

$$u = 0,1,...,M-1; v = 0,1,...,N-1$$

Among them, $C(u) = \begin{cases} \sqrt{\dfrac{1}{M}}, u=0 \\ \sqrt{\dfrac{2}{M}}, u \neq 0 \end{cases}$ , $C(v) = \begin{cases} \sqrt{\dfrac{1}{N}}, v=0 \\ \sqrt{\dfrac{2}{N}}, v \neq 0 \end{cases}$ , $f(x,y)$ is the pixel value of

point $(x,y)$, and $F(u,v)$ is the 2D-DCT transform coefficient of $f(x,y)$. DCT is preferred in digital image processing as compared to other transformations, such as DFT or FFT, because signal will "lose its form" if the representation coefficients are truncated in DFT because the signal is represented periodically. Due to the continuous periodic structure in DCT, however, the signal can withstand larger amounts of coefficient truncation while still maintaining the desired shape [29].

### 2.4. Logistic Map

The logistic map is one of the most famous chaotic mappings, a simple dynamic non-linear regression with chaotic behavior, as shown in Figure 3 [30]. Its mathematical definition can be expressed as follows:

$$X_{k+1} = \mu \cdot X_k \cdot (1 - X_k) \tag{4}$$

Among them, $X_k \in (0,1), 0 < \mu \leq 4$.

Experiments show that when $3.5699456 < \mu \leq 4$, the logistic mapping enters a chaotic state and the logistic chaotic sequence can be used as an ideal key sequence.

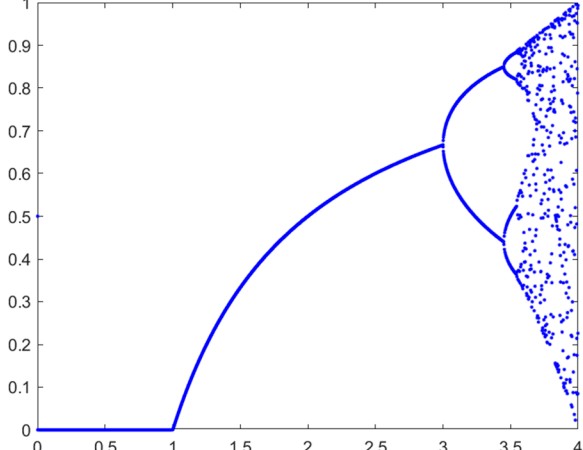

**Figure 3.** Logistic mapping bifurcation diagram [31].

## 3. The Proposed Algorithm

In this paper, we propose a robust watermarking algorithm for encrypted medical images based on DCT and an improved DarkNet53 convolutional neural network [32]. The main parts are: improvement of the network, migration learning, encryption of medical images and watermark information, feature extraction of images and embedding, and extraction of watermarks.

### 3.1. Medical Image Encryption

This paper opts to embed and remove the watermark in the ciphertext domain due to the unique nature of medical photographs, and the corresponding encryption method is depicted in Figure 4. Firstly, the coefficient matrix $D(i, j)$ is obtained by DCT of the original image; secondly, the tent mapping chaotic sequence $X(j)$ is extracted and binarized to obtain $C(i, j)$; and finally, the dot product operation is performed on $D(i, j)$ and $C(i, j)$ to obtain $ED'(i, j)$. Finally, the IDCT transform of $ED'(i, j)$ is performed to obtain the encrypted medical image.

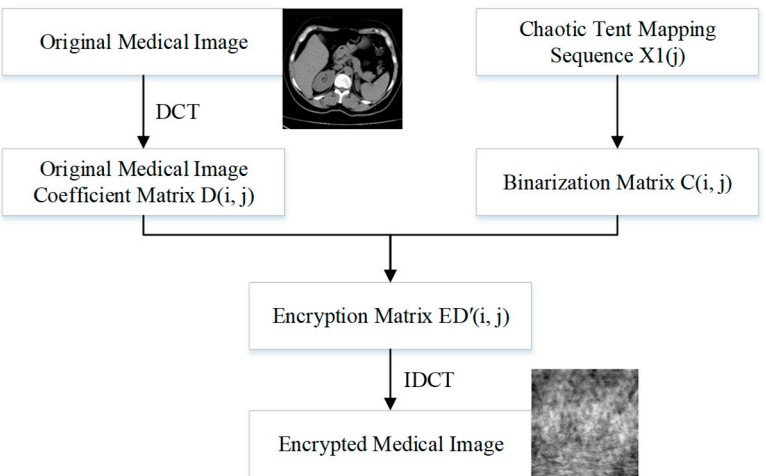

**Figure 4.** Image encryption process.

### 3.2. Improved DarkNet53 Network Model

3.2.1. Improvement of Network Structure

Since convolutional neural networks have powerful feature extraction abilities and can effectively extract stable features of images, they are of high research value for medical image watermarking algorithms. In this paper, the DarkNet53 convolutional neural network, which has been trained on more than one million natural images in the ImageNet database, is selected for its inherently good feature extraction ability. In order to achieve strong robustness in the watermarking algorithm, this paper makes a simple change to the network. First, the Softmax layer and classification layer of the DarkNet53 convolutional neural network are removed, then a fully connected layer with 128 values of output is added, and finally a regression output layer is connected after the fully connected layer so that the modified network can complete the regression task and the output of the fully connected layer is used as our extracted medical image features.

3.2.2. Data Set Creation

The medical image data selected for this paper comes from the Medical Imaging Park and American Research Institute, Inc., which contains tens of thousands of medical images. In this paper, we selected 125 medical images in each of the five major categories of brain, pelvis, bone and muscle, colon, and chest from the website as dataset 1, and encrypted these 125 medical images to obtain 125 encrypted medical images as dataset 2.

Some of these 250 images are shown in Figure 5. Data sets 1 and 2 are completely disrupted, respectively, and then divided into three parts in the ratio of 3:1:1: the training set, validation set, and test set. In order to improve the robustness of the network in extracting features, in this paper, the training set and the validation set are enhanced with the data separately, as shown in Table 1. Thus, we obtained 12,850 images as the total dataset for this training. Because the input size of the DarkNet53 convolutional neural network is 2,562,563, all medical images are resized to 2,562,563 here. For the production of dataset labels, this paper first performs DCT on the images of the training and validation sets and then selects the 128-bit feature vectors of the low-frequency part as labels.

**Table 1.** Specific implementation operations for data enhancement.

| Enhancement Methods | Intensity | Number of New Images |
|---|---|---|
| Gaussian noise (%) | 3, 6, 9, 12, 15 | 5 |
| JPEG compression (%) | 5, 10, 15, 20, 25 | 5 |
| Median filter (10 times) | 3 × 3, 5 × 5, 7 × 7 | 3 |
| Clockwise rotation (°) | 5, 10, 15, 20, 25, 30, 35, 40 | 8 |
| Scaling | 0.3, 0.6, 0.9, 1.2, 1.5, 1.8 | 6 |
| Down-shift (%) | 5, 10, 15, 20, 25, 30 | 6 |
| Up-shift (%) | 5, 10, 15, 20, 25, 30 | 6 |
| Y-axis shear (%) | 5, 10, 15, 20, 25, 30 | 6 |
| X-axis shear (%) | 5, 10, 15, 20, 25, 30 | 6 |
| Left-shift (%) | 5, 10, 15, 20, 25, 30 | 6 |
| Right-shift (%) | 5, 10, 15, 20, 25, 30 | 6 |

### 3.2.3. Training Network

The computer configuration used for this experiment was an NVIDIA GeForce GTX 1050Ti 4GB graphics card (Santa Clara, CA, USA) and Intel@RCoreTM i5-8300H CPU @ 2.30GHz4 (Santa Clara, CA, USA). The software uses the neural network toolbox that comes with Matlab 2022a, and the network selected was the DarkNet53 pre-trained network. In this paper, the DarkNet53 pre-trained network is trained by first setting the learning rate of 1:84 layers to 0 to "freeze" the weights of these layers, because the parameters of the frozen layers will not be updated during the whole training process. Next, the initial learning rate is set to 0.001, the MiniBatchSize is set to 30, and the Epochs are set to 8 for training. Finally, the trained network is saved as a key part of the watermarking algorithm.

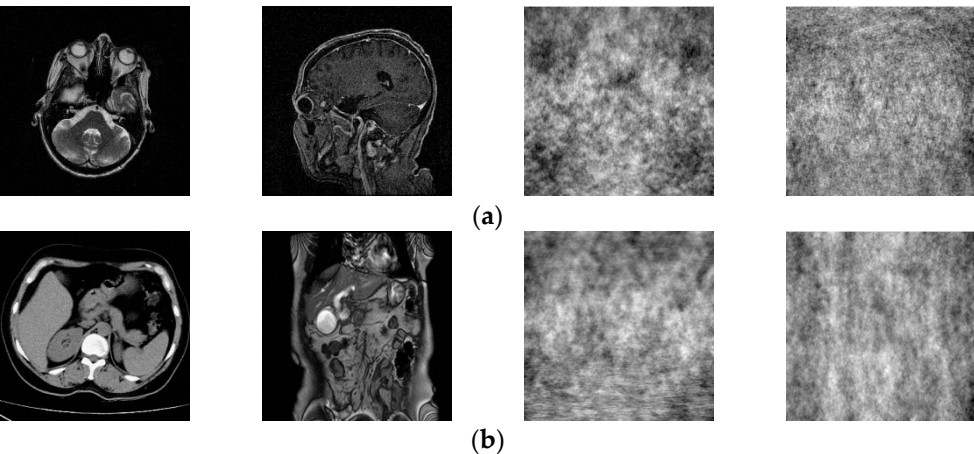

(a)

(b)

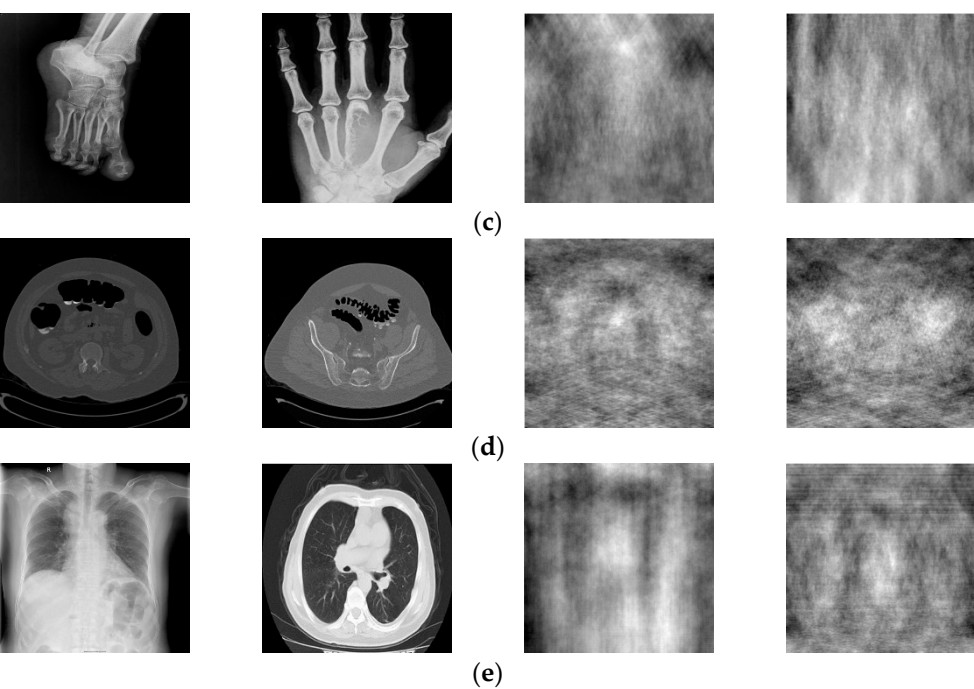

**Figure 5.** Two original images and corresponding encrypted images for each type of image. Brain image (**a**) ; abdominal pelvis image (**b**); bone muscle image (**c**); colon image (**d**); and chest image (**e**).

*3.3. Encryption of Watermarks*

In this research, we make use of a chaotic system of logistic mapping to create an encrypted watermarked image through the utilization of chaotic dislocation. The watermarking process will now be more secure and resistant to interference as a result of this change. The first thing that needs to be done in order to generate the encrypted watermarked image is to enter the chaotic system by providing values for the coefficients and the initial state. This is done so that the system can begin to generate the image. To begin the process of decrypting the image, we will first need to use the chaotic system to create a chaotic sequence. After we have obtained this sequence, we will need to perform a bit-by-bit XOR operation between it and the binary image that has been watermarked with it. The intricate algorithmic structure of the watermark chaos encryption is shown in Figure 6.

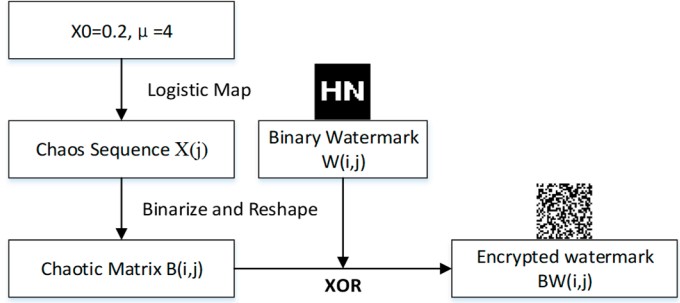

**Figure 6.** Watermark encryption process.

*3.4. Feature Extraction of Encrypted Medical Images*

Traditional watermarking algorithms mainly embed watermarks directly into medical images, which may not only change the quality of the images but also their resistance to attacks, especially geometric attacks, which often do not have good robustness. In this paper, we perform DCT on encrypted medical images and select the 32-bit feature vector

of the low-frequency part for hash transform as feature 1, denoted as V1(i, j) [33]. Meanwhile, the 128-bit feature matrix of the fully connected layer is extracted from the encrypted medical image using a DarkNet53 convolutional neural network after migration learning, and then DCT is performed on this 128-bit feature matrix, and the 32-bit feature vector of the low-frequency part is extracted for the hash transform as feature 2, denoted as V2(i, j). Finally, the feature set V(i, j) is established, which can be better combined with the zero-watermarking technique. The specific steps are shown in Figure 7.

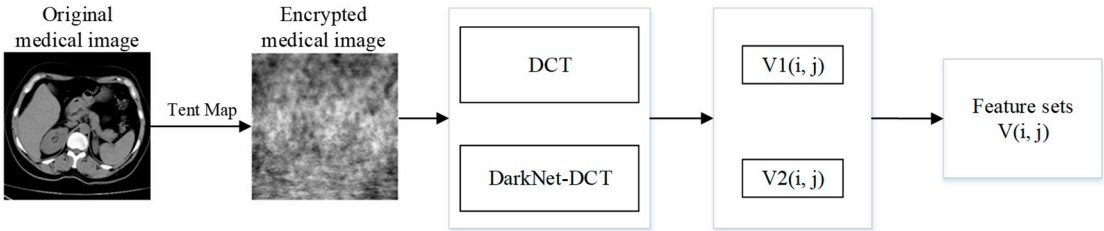

**Figure 7.** Extraction process of encrypted medical image features.

### 3.5. Encrypted Watermark Embedding

The process of watermark embedding in this paper is primarily broken down into the following steps: the feature vectors V1(i, j) and V2(i, j) in the feature set and each row of the encrypted watermark BW(i, j) are each subjected to bit-by-bit XOR operations, respectively; the watermark information can then be hidden in the medical image. Additionally, when embedding a watermark using this method, there is no change to the pixel values in the original image, achieving a zero watermark. The specific watermark embedding process is shown in Figure 8.

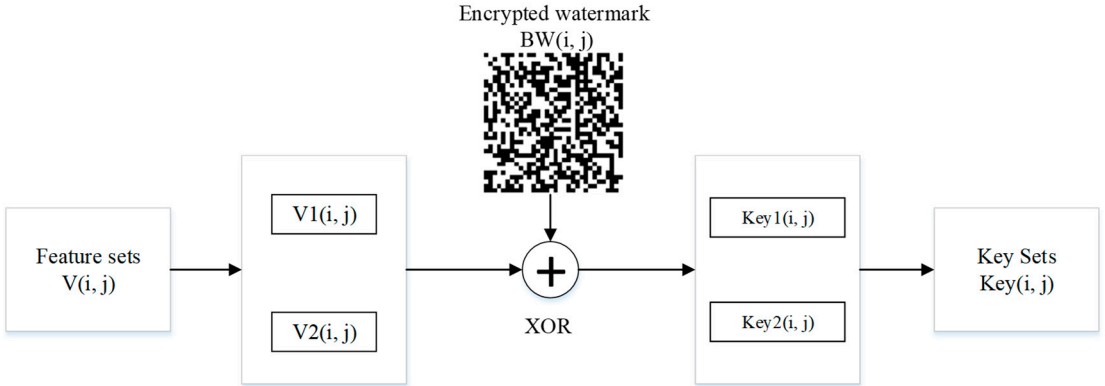

**Figure 8.** Flow chart of embedding watermark.

### 3.6. Extraction of Watermarks

The steps for watermark extraction are identical to those for the watermark embedding process, where V1'(i, j) is the feature vector extracted by DCT and V2'(i, j) is the feature vector extracted by DarkNet-DCT, and the specific operation flow is shown in Figure 9.

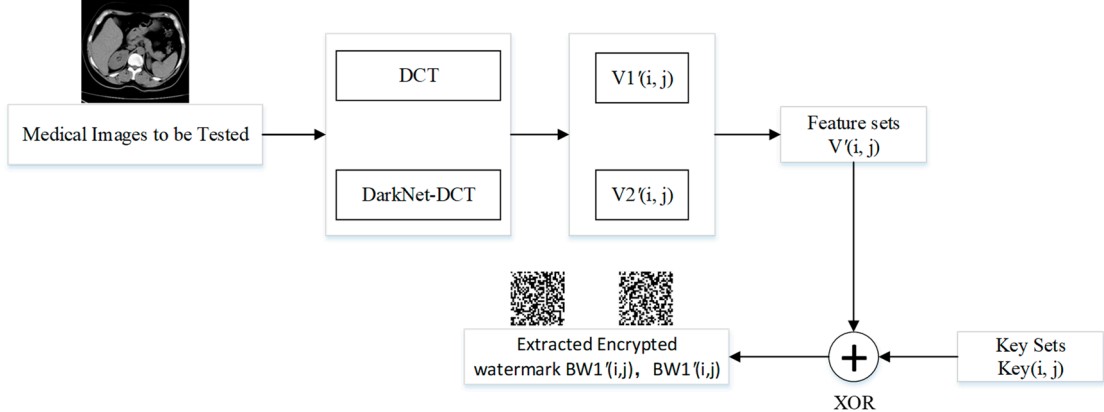

**Figure 9.** Flowchart of watermark extraction.

*3.7. Decryption of Watermark*

The decryption of the watermark is consistent with the chaotic sequence X(j) used in the encryption method of the watermark, and X(j) and the encrypted watermarks BW1′(i, j) and BW2′(i, j) are subjected to the XOR operation to obtain the decrypted watermarks W1′(i, j) and W2′(i, j), respectively. Calculate the correlation coefficients NC1 and NC2 of W(i, j) and W′(i, j), then discriminate NC1 and NC2, and output the larger correlation coefficient and the corresponding watermark image. The specific flow of watermark extraction and decryption is shown in Figure 10.

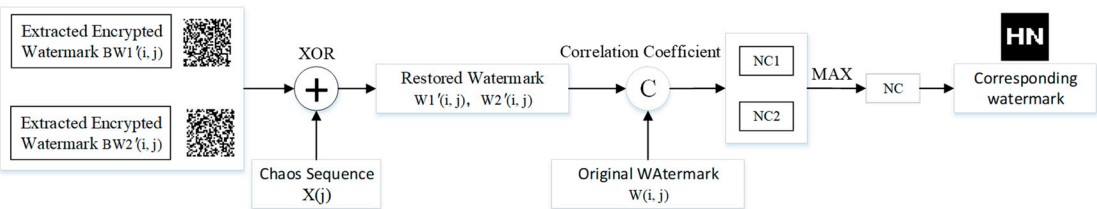

**Figure 10.** Decryption process of watermark.

**4. Experimental Results and Analysis**

In this research, a medical image and a 32 × 32 watermarked image carrying information from the test set were randomly selected for the Matlab 2022a simulation platform to study and test the robustness of the mentioned watermarking algorithm. To improve the security, we used tent chaos mapping to encrypt the medical image and logistic chaos encryption to encrypt the watermarked image, as shown in Figure 11.

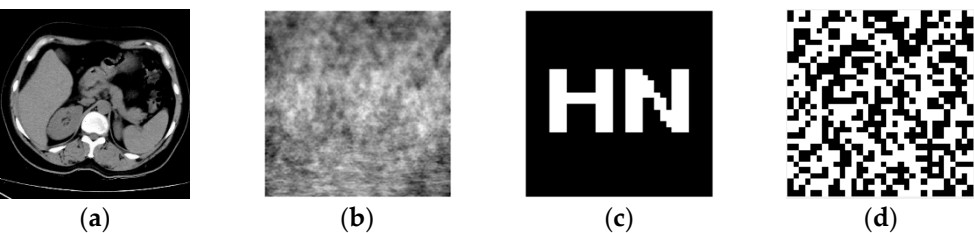

| (**a**) | (**b**) | (**c**) | (**d**) |

**Figure 11.** Medical image and watermarked image. Original medical image (**a**), encrypted medical image (**b**), original watermarked image (**c**), and encrypted watermarked image (**d**).

*4.1. Performance Index*

In this paper, the robustness of the algorithm is reflected by the normalized correlation coefficient (NC) and the peak signal-to-noise ratio (PSNR). Where NC indicates the

similarity between the original watermark and the extracted watermark, and the closer its value is to 1, the higher the correlation between the two and the better the robustness of the algorithm. The calculation formula is shown in (5). PSNR indicates the degree of distortion of the image containing the watermark, and a smaller value means a greater degree of distortion of the original image; the calculation formula is shown in Equation (6).

$$NC = \frac{\sum_i \sum_j W_{(i,j)} W_{(i,j)}'}{\sum_i \sum_j W_{(i,j)}^2} \tag{5}$$

$$PSNR = 10 \lg \left[ \frac{MN \max_{i,j} \left( I(i,j) \right)^2}{\sum_i \sum_j \left( I(i,j) - I'(i,j) \right)^2} \right] \tag{6}$$

### 4.2. Reliability Analysis

In order to prove that the deep-learning algorithm proposed in this paper has certain reliability, eight medical images were randomly selected from the test set for testing, as shown in Figure 12. The NC is used to calculate the correlation between different images, and when the NC < 0.5, it indicates that the correlation between different image feature vectors is low, and the feature vectors extracted by this algorithm are representative. Table 2 shows the NC values between eight different cryptographic medical images. Since the absolute values of NC values of different images are less than 0.5 and the NC value of the same image is 1, the algorithm can distinguish different encrypted medical images and is reliable.

**Table 2.** NC between different encrypted images.

| Image | 1 | 2 | 3 | 4 | 5 | 6 | 7 | 8 |
|---|---|---|---|---|---|---|---|---|
| 1 | 1 | | | | | | | |
| 2 | 0.32 | 1 | | | | | | |
| 3 | 0.22 | 0.22 | 1 | | | | | |
| 4 | 0.26 | 0.04 | 0.12 | 1 | | | | |
| 5 | 0.25 | 0.07 | 0.17 | 0.24 | 1 | | | |
| 6 | 0.01 | 0.16 | 0.01 | 0.22 | 0.49 | 1 | | |
| 7 | 0.12 | 0.07 | 0.17 | 0.11 | 0.12 | 0.24 | 1 | |
| 8 | 0.37 | 0.05 | 0.36 | 0.39 | 0.25 | 0.11 | 0.25 | 1 |

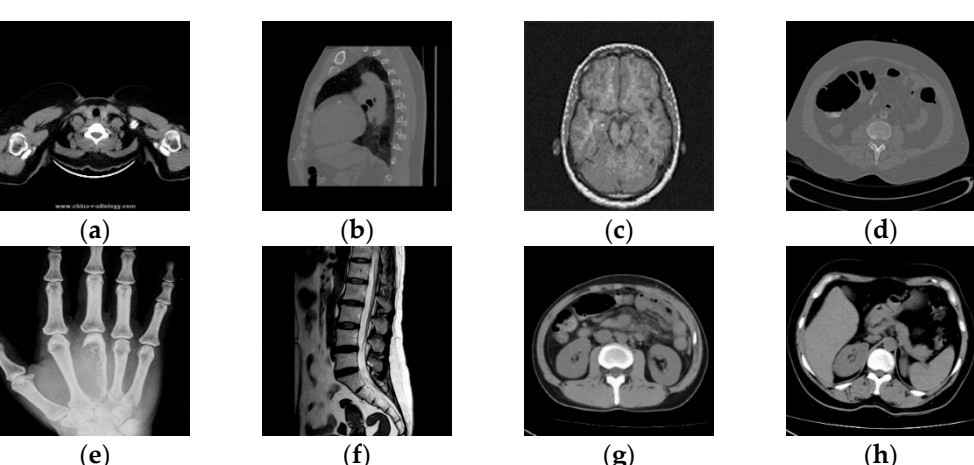

(a)    (b)    (c)    (d)

(e)    (f)    (g)    (h)

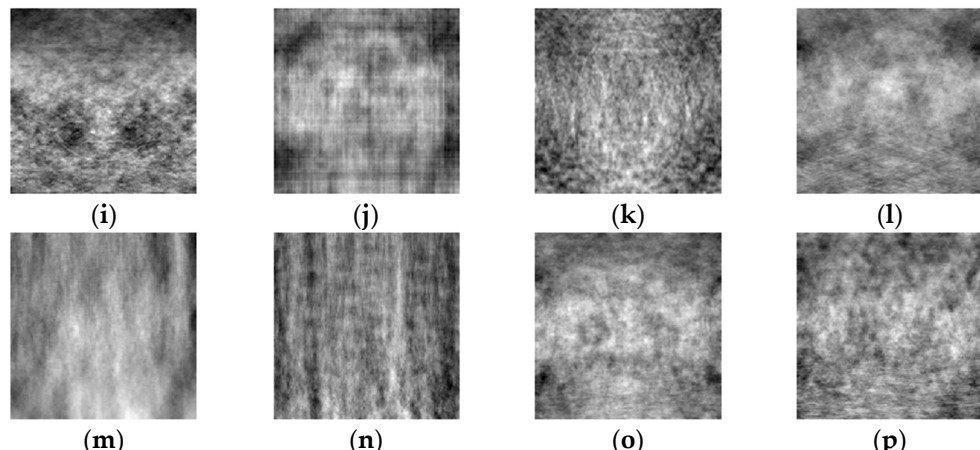

**Figure 12.** Different medical images within the test (**a–h**), related encrypted images (**i–p**).

### 4.3. Conventional Attacks

To test the robustness of the algorithm against conventional attacks, three conventional attacks—Gaussian noise, JPEG compression, and median filtering—were selected for testing in this paper. The NC value between the original watermark and the extracted watermark was calculated, with larger values indicating better robustness. The experimental results are shown in Table 3. It can be seen that when the Gaussian attack reaches 13%, the NC value is 1.00; when the JPEG compression quality is 5%, the NC value is 0.85; and when the median filtering [7 × 7] is 10 times, the NC value is 1.00. Figure 13 shows some of the experimental results. It shows that the watermark information can be effectively recovered with good robustness in the face of all three conventional attacks.

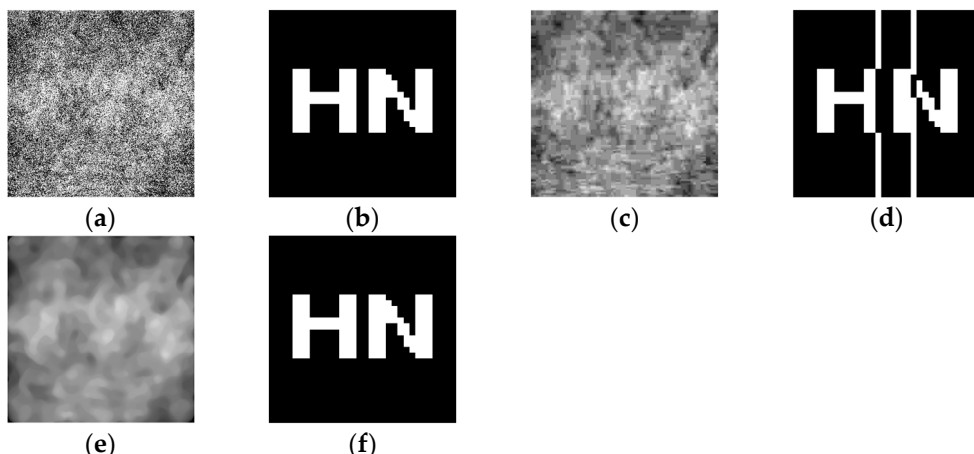

**Figure 13.** Encrypted medical images and extracted watermarks after some conventional attacks. Images after 13% attack of Gaussian noise and corresponding extracted watermarks (**a**,**b**), images after 5% attack of JPEG compression and corresponding extracted watermarks (**c**,**d**), and images after 10 attacks of median filtering [7 × 7] and corresponding extracted watermarks (**e**,**f**).

**Table 3.** Experimental data under conventional attack.

| Attacks | Intensity | PSNR(dB) | NC |
|---|---|---|---|
| | 3 | 15.37 | 0.91 |
| | 7 | 12.27 | 1.00 |
| Gaussian noise (%) | 9 | 11.47 | 0.94 |
| | 13 | 10.43 | 1.00 |
| | 5 | 26.46 | 0.85 |
| JPEG compression (%) | 15 | 31.36 | 0.91 |
| | 30 | 34.32 | 1.00 |

| | | | |
|---|---|---|---|
| | [3 × 3] | 31.35 | 1.00 |
| Median filter | [5 × 5] | 26.10 | 1.00 |
| | [7 × 7] | 23.95 | 1.00 |

### 4.4. Geometric Attacks

As geometric attacks are a more difficult problem for existing algorithms to solve, this paper tests the robustness of the algorithm after rotation, scaling, and shear attacks. The NC value between the original watermark and the extracted watermark was calculated, with larger values indicating that the algorithm is more resistant to geometric attacks. The experimental results are shown in Table 4. It can be seen that the NC value was 0.62 when rotating 30° counterclockwise. When rotating 50° clockwise, the NC value was 0.53. When the scaling factor was between 0.1 and 2, the NC value was 1. When shifting 30% left, the NC value was 0.62. When shifting 40% right, the NC value was 0.64. When shifting 40% up, the NC value was 0.72. When shifting 40% down, the NC value was 0.58. When shearing 40% along the X-axis, the NC value was 0.55. When shearing 40% along the Y-axis, the NC value was 0.67. When shearing 40% along the X-axis direction, the NC was 0.55. When shearing 40% along the Y-axis direction, the NC was 0.67. The watermark information could be effectively recovered for all the above geometric attacks. Figure 14 shows the experimental results after partial geometric attacks.

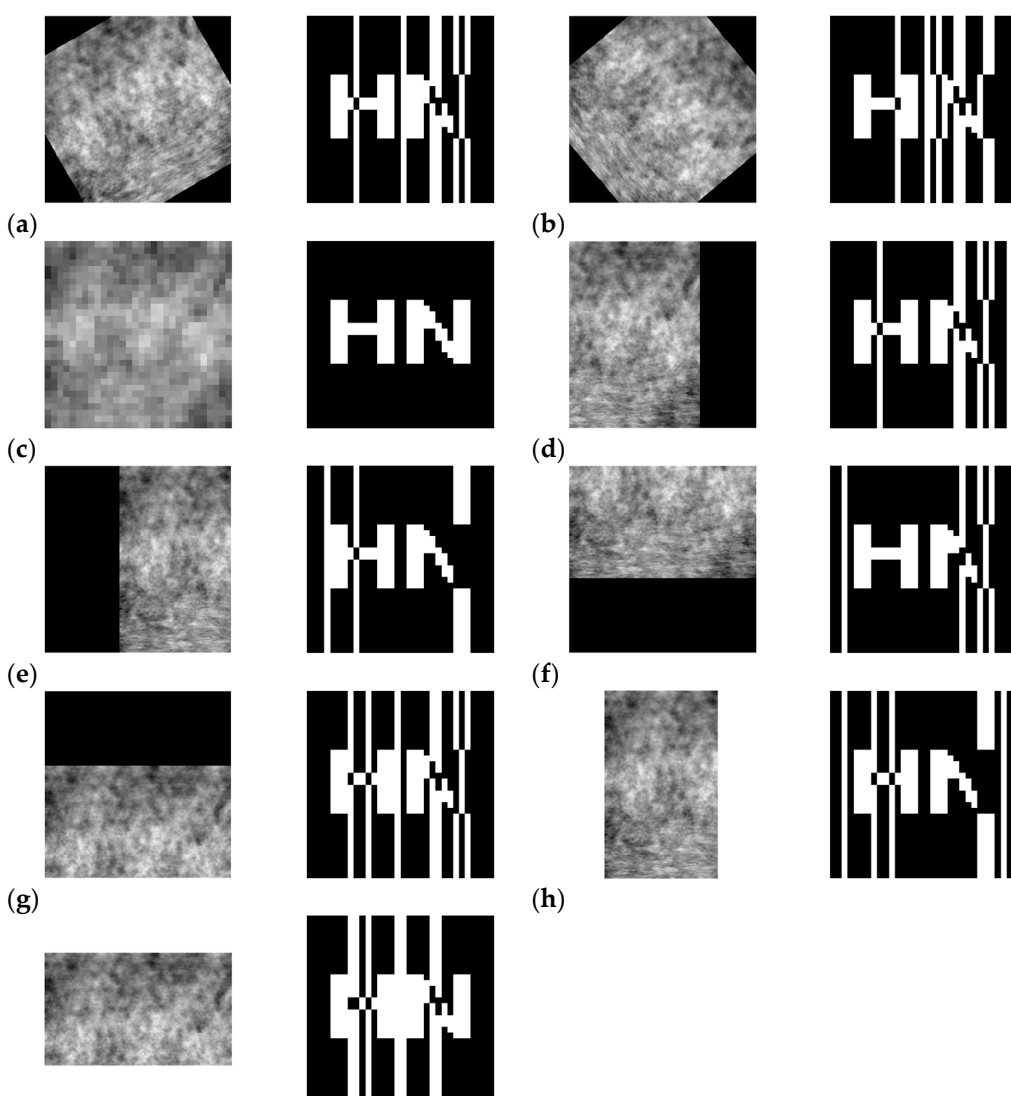

(**a**)　　　　　　　　　　　　(**b**)

(**c**)　　　　　　　　　　　　(**d**)

(**e**)　　　　　　　　　　　　(**f**)

(**g**)　　　　　　　　　　　　(**h**)

(**i**)

**Figure 14.** Encrypted medical images and extracted watermarks after some geometric attacks. Image and extracted watermark after 30° counterclockwise rotation (**a**), image and extracted watermark after 50° clockwise rotation (**b**), image and extracted watermark after 0.1 times scaling (**c**), image and extracted watermark after 30% left-shift (**d**), image and extracted watermark after 40% right-shift (**e**), image and extracted watermark after 40% up-shift (**f**), image and extracted watermark after 40% down-shift (**g**), image and extracted watermark after 40% clipping in X-axis direction (**h**), and image and extracted watermark after 40% clipping in Y-axis direction watermark (**i**).

**Table 4.** Experimental data under geometric attack.

| Attacks | Intensity | PSNR(dB) | NC |
|---|---|---|---|
| | 10 | 15.43 | 0.78 |
| Anticlockwise Rotation (°) | 20 | 13.90 | 0.62 |
| | 30 | 13.16 | 0.62 |
| | 10 | 15.14 | 0.85 |
| Clockwise Rotation (°) | 30 | 13.21 | 0.80 |
| | 50 | 12.73 | 0.53 |
| | 0.1 | - | 1.00 |
| Scaling Factor | 0.5 | - | 1.00 |
| | 2 | - | 1.00 |
| | 10 | 13.32 | 0.81 |
| Translation Left (%) | 20 | 11.01 | 0.63 |
| | 30 | 9.64 | 0.62 |
| | 5 | 16.16 | 0.80 |
| Translation Right (%) | 20 | 11.57 | 0.81 |
| | 40 | 8.90 | 0.64 |
| | 10 | 13.28 | 0.85 |
| Translation Up (%) | 20 | 10.92 | 0.75 |
| | 40 | 8.46 | 0.72 |
| | 5 | 15.94 | 0.86 |
| Translation Down (%) | 20 | 11.82 | 0.81 |
| | 40 | 8.84 | 0.58 |
| | 10 | - | 0.85 |
| X-axis Crop (%) | 20 | - | 0.87 |
| | 40 | - | 0.55 |
| | 10 | - | 0.85 |
| Y-axis Crop (%) | 20 | - | 0.76 |
| | 40 | - | 0.67 |

*4.5. Comparison between Different Algorithms*

In order to better verify the robustness of this algorithm, this paper compares the more classical watermarking algorithms, Inception V3-DCT, PHTs-DCT, KAZE-DCT, DWT-DCT, and Curvelet-DCT, in recent years [34–38]. During the experiments, the same medical image and watermarked image were selected for testing in order to ensure the consistency of the conclusions. The results of the comparison experiments are shown in Table 5 and Figure 15. It can be seen that DWT-DCT and Curvelet-DCT have the best results in the face of traditional attacks. Facing the geometric attack, the algorithm proposed in this paper shows strong robustness. In a comprehensive comparison, the algorithm proposed in this paper demonstrates stronger robustness in the face of different geometric attacks and conventional attacks.

**Table 5.** Comparison of NC values between different algorithms.

| Attacks | Intensity | Inception V3-DCT [34] | PHTs-DCT [35] | KAZE-DCT [36] | DWT-DCT [37] | Curvelet-DCT [38] | Proposed |
|---|---|---|---|---|---|---|---|
| Gussian Noise | 13 | 0.35 | 0.45 | 0.32 | **1.00** | **1.00** | 0.94 |
| JPEG Compression | 10 | 0.63 | 0.63 | 0.76 | **1.00** | **1.00** | 0.94 |
| Median Filter | [7 × 7] | 0.29 | 0.55 | 0.40 | 0.84 | **1.00** | **1.00** |
| Rotation (°) | 30 | 0.05 | 0.62 | 0.42 | 0.46 | 0.41 | **0.80** |
| Scaling | ×0.1 | 0.32 | - | 0.43 | 0.90 | 0.90 | **1.00** |
| Right Translation (%) | 40 | 0.39 | 0.49 | 0.04 | 0.13 | 0.20 | **0.64** |
| Up Translation (%) | 40 | 0.39 | 0.31 | 0.20 | 0.02 | 0.02 | **0.72** |
| Cropping (X-axis) | 20 | 0.76 | 0.59 | 0.68 | 0.31 | 0.30 | **0.87** |
| Cropping (Y-axis) | 15 | 0.68 | 0.45 | 0.62 | 0.65 | 0.74 | **0.81** |

Note: The bold part indicates that the algorithm has the best robustness compared to these three algorithms.

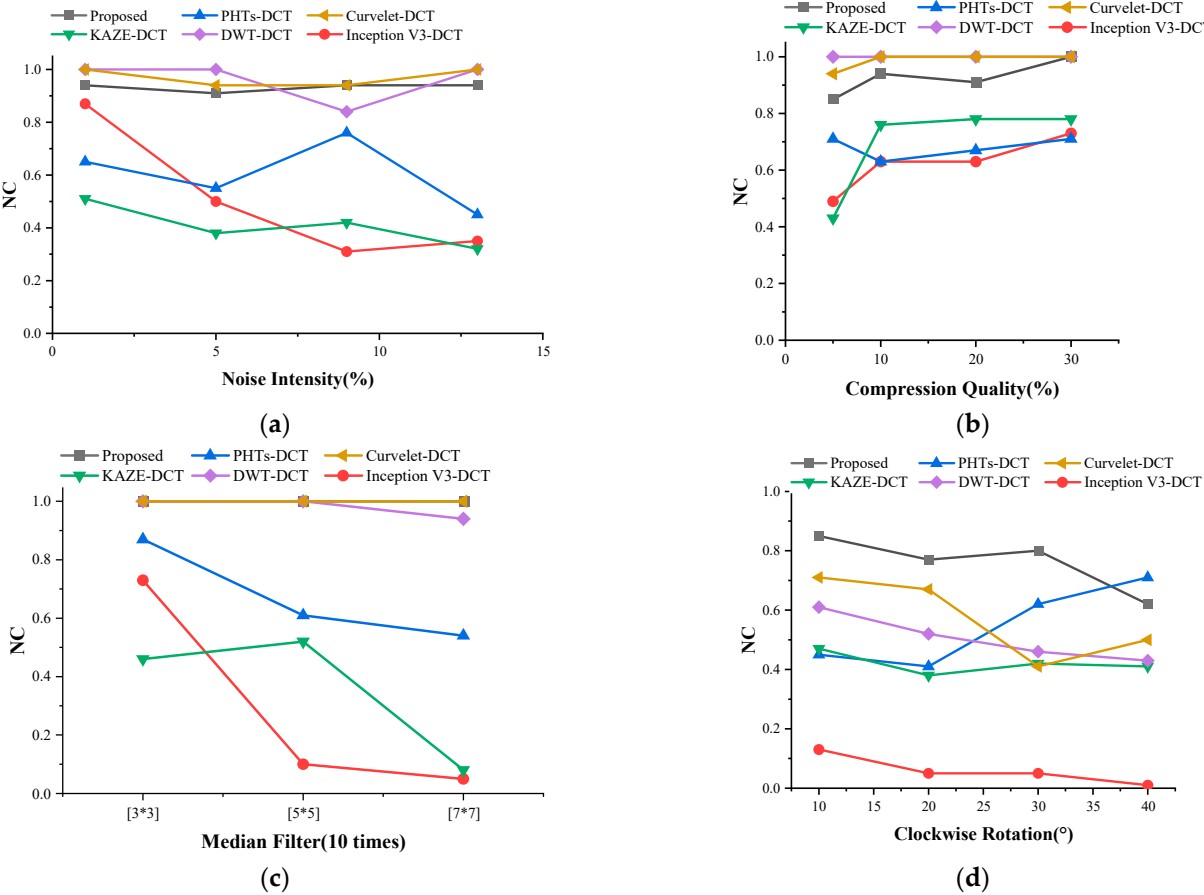

(a)

(b)

(c)

(d)

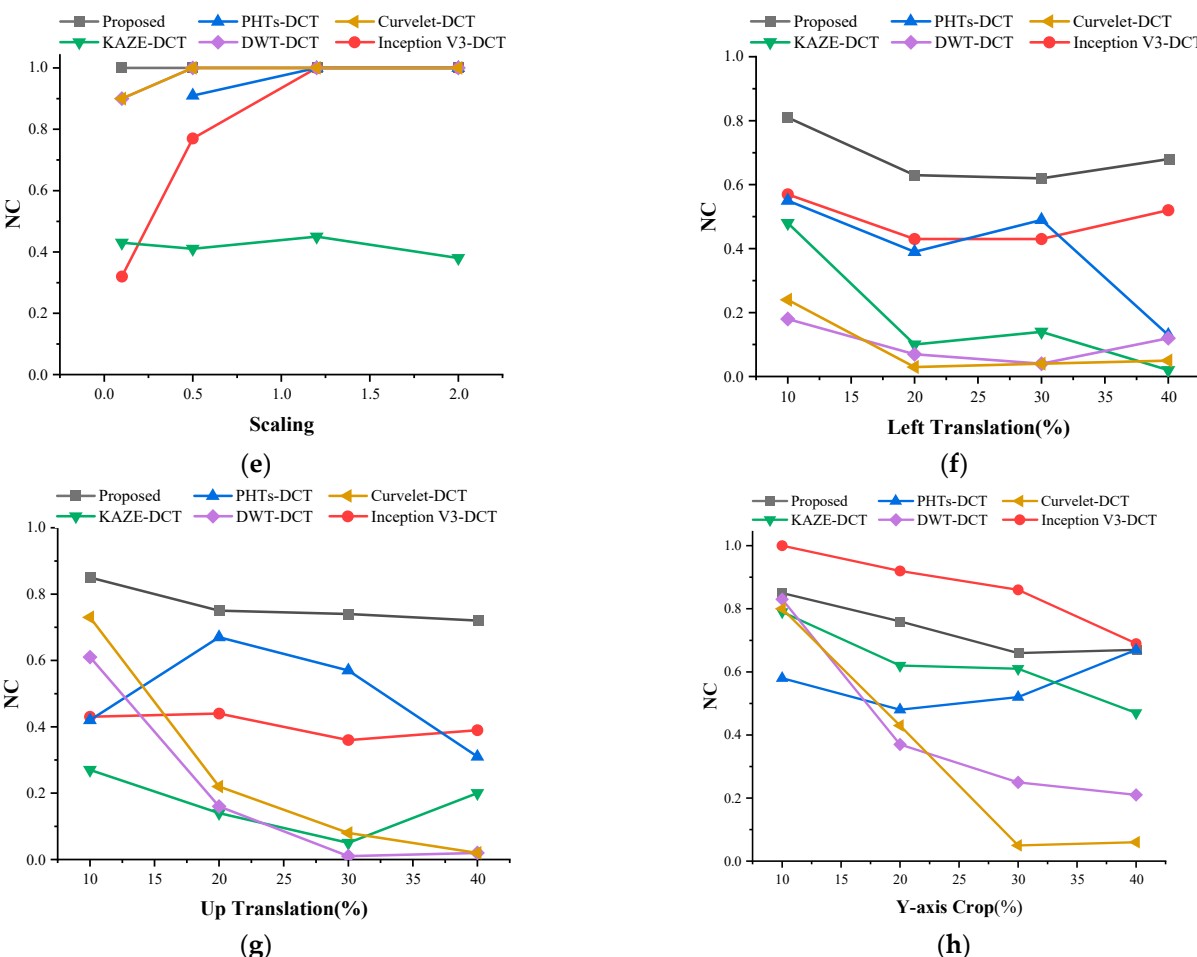

**Figure 15.** Comparison of NC values between different algorithms, where (**a**–**h**) indicate the results after Gaussian noise, JPEG compression, median filter, clockwise rotation, scaling, left-shift, up-shift, and Y-axis shear attacks, respectively.

## 5. Conclusions

In this paper, we propose an encrypted medical image watermarking algorithm that is based on DCT and Darknet53 convolutional neural networks. This algorithm combines migration learning, DCT, Tent Map, Logistic Map, hash transform, and zero watermarking techniques. The algorithm is designed to protect medical images from unauthorized use. Improving the Darknet53 pre-trained network was the first step in the experimental procedure. Next, migration learning was performed on the improved network in order to extract features from encrypted medical images. After that, the medical images and watermark information were encrypted with Tent Map and Logistic Map. Finally, the zero-watermarking technique was used to embed the watermark information and then extract it. The findings of the experiments demonstrate that the method has a high degree of robustness when subjected to both conventional and geometric attacks. As a result, the technique may prove to be superior for use with encrypted medical photos. Naturally, there is still a great deal of space for development of this method, and in order to enhance the performance of the algorithm, we will continue to tune the neural network in order to extract characteristics that are more reflective of the whole.

**Author Contributions:** Formal analysis, validation, data curation, and writing—original draft preparation, D.L.; funding acquisition, J.L. (Jingbing Li); visualization, S.A.N.; supervision, J.L. (Jing Liu); software, Y.-W.C.; investigation, L.C. Data curation, Supervision , Resources, Writing—original draft, U.A.B. All authors have read and agreed to the published version of the manuscript;

**Funding:** This research was supported, in part, by the Natural Science Foundation of China under grants 62063004, the Key Research Project of Hainan Province under grant ZDYF2021SHFZ093, the Hainan Provincial Natural Science Foundation of China under grants 2019RC018 and 619QN246, and the postdoctoral research from Zhejiang Province under grant ZJ2021028.

**Data Availability Statement:** Data is contained within the article.

**Conflicts of Interest:** The authors declare no conflict of interest.

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
