# Peer review of "Hybrid Encrypted Watermarking Algorithm for Medical Images Based on DCT and Improved DarkNet53"

_electronics, doi:10.3390/electronics12071554_

Round 1

Reviewer 1 Report

In this work, the authors have reported a watermarking approach for medical images using DCT and Darknet53. The paper should be revised based on the following suggestions.

1. Abstract should highlight the underlying issues and the results.

2. Why does the DCT only used? any benefits over DFT or any other?

3. Introduction should extensively present the benefit and issues of recent watermarking approaches such as, https://link.springer.com/article/10.1007/s42979-022-01657-1 and https://www.mdpi.com/2079-9292/12/5/1222

4. why did the tent map bifurcation is used? Is it to randomize the secret bits? In this, should the authors not use the chaotic map suggested in https://ieeexplore.ieee.org/abstract/document/9999446

5. section 2.4 should discuss in-depth the logistic -map considering the work suggested in the before comment.

6. What is the source of the dataset and hoe many images are present in it should be mentioned?

7. Present an illustration of the encryption and decryption process.

8. Experimental result is well explained.

Reviewer 2 Report

In this paper, the authors proposed that DCT and an improved DarkNet53 convolutional neural network can make a robust zero-watermarking algorithm for cryptographic medical images. The security and appropriate watermarking of medical images are essential because medical images contain sensitive data, and the patients' data should be secure.
This article is well-organised and written, and I suggest it for publication.

Author Response

请参阅附件。

Reviewer 3 Report

This work combines the discrete cosine transform with an enhanced DarkNet53 convolutional neural network for zero watermarking of medical images. The manuscript is generally well prepared with some interesting results. Following are some comments that need to be considered:

1- Regarding the organization of the paper, I would propose reorganizing and enhancing the introduction part so as to distinguish the related work from the remaining points. The introduction should also explain why DarkNet53 is used in this context and justify its ability to deal with the zero watermarking of medical images when combined with DCT. In addition, related work should contain surveys or review papers on the issue of medical image zero-watermarking, as well as a table summarizing the related works cited.

2-Lines 97-101 are supposed to describe the work proposed by authors in reference [20] where VGG-19 deep convolutional neural network is used. The use of the pronoun "We" suggests that the authors own this work which is not true. This point needs clarification and correction.

3-Some figures (2 and 3) require proper citation and maybe permission to reuse to avoid copyright issues.

4-Figure 1 contains a typo (see the word convlutional).

5-Citations should be properly attached to equations.

6-In line 170 citation [26] is mentioned while the paragraph describes the proposed approach. Why such a citation is given at this point? Authors should clarify the relation of their work with the work in [26].

7-The authors refer to feature vectors using notations such as in lines 254-255 V1(i,j), V2(i,j), BW (i,j). Usually, such notations relate to matrix elements. I would recommend reconsidering the current notation with a clear and detailed description of each symbol.  It should be clear what the indexes i and j refer to.

8-The source of the data sets used is mentioned. However, there is no reference to this source.

9-Whenever the performance metric NC is used on pages 12 and 13, it should be explicitly stated between which pair of images it is computed. For instance, on page 13 it is calculated between the original watermark and the extracted one which explains that higher values are desirable (table 3) contrary to table 2.

10-In table 5, according to [32], the method combines curvelet-DCT and RSA. This should be reported in the table.

11-Finally, and most importantly, the paper does not state whether the results shown are the average of results over the testing images. Is it the case? The methods used in the comparative study must be properly compared using statistical tests to determine whether the difference is statistically significant. Boxplots may also provide additional information to support the author's results and conclusions. Do the methods in this comparative study use the same images? How can you ensure that the comparative study is unbiased?

Author Response

请参阅附件。

Round 2

Reviewer 1 Report

The authors have successfully revised the manuscript

Author Response

thanks for accepting manuscript

Reviewer 3 Report

Despite the fact that the authors did not adequately respond to comment 11, the paper is still acceptable in its current form. Minor text editing is required for the paper.

Author Response

thank you for accepting our manuscript.
our manuscript is checked by foreign countries authors and believe that manuscript is acceptable.